# What happened and what proves you wrong? Combatting confirmation bias in police investigations through evidence reconstruction and falsification

Sarah Lenz[1], Tara Zohrevand[1], Eric Rassin[2], Bruno Verschuere[1]*

**1** University of Amsterdam, Amsterdam, The Netherlands, **2** Erasmus University, Rotterdam, The Netherlands

befThese authors contributed equally to this work.
* b.j.verschuere@uva.nl

## Abstract

Confirmation bias in criminal investigations has repeatedly been linked to wrongful convictions. Drawing on principles from the Scenario Reconstruction Method, developed for and used in Dutch policing, this study tested whether shifting the investigative focus from identifying suspects to reconstructing scenarios based on available evidence could reduce confirmation bias. In addition, the design included a theoretically motivated manipulation encouraging falsification over verification. The result was a 2 (focus: suspect vs. evidence) × 2 (strategy: verification vs. falsification) factorial online study involving 293 current and future German police officers, who analysed a real wrongful conviction case from the Netherlands. The primary outcome was the accuracy of guilt ratings for the innocent suspect; secondary outcomes assessed the type of proposed next investigative steps. The analyses showed the manipulations had no effect on guilt ratings. However, both strategies did influence investigative reasoning: evidence-focus increased the likelihood of proposing more evidence-based next investigative steps, while falsification-focus promoted more falsification-oriented next investigative steps. Cross-over effects suggested a broader shift in investigative mindset toward more objective reasoning. Future research should explore whether these early improvements in reasoning translate into more accurate outcomes when progressing from brief instructions to multi-stage interventions such as the full Scenario Reconstruction Method.

## Introduction

In June 2000, two children were attacked in a park in Schiedam [1]. The perpetrator attempted to sexually assault both children before trying to kill them. One of the children survived by playing dead. The police soon focused on Kees B, a man with a

**Data availability statement:** All files (data, analysis scripts and materials) are available on FigShare: https://figshare.com/s/9665c4888a17e48636e1.

**Funding:** The author(s) received no specific funding for this work.

**Competing interests:** The authors have declared that no competing interests exist.

known history of paedophilic behaviour, who had previously made sexual advances toward another boy in the same park and went to the park again around the time of the crime for the same purpose. Despite clear discrepancies between the surviving child's detailed description of the perpetrator and Kees B's appearance, as well as tachograph data confirming that Kees B could not have arrived at the crime scene in time to commit the murder, Kees B was convicted. Years later, the real perpetrator confessed to the crime, and DNA evidence confirmed his guilt. Kees B was exonerated, but only after years of wrongful imprisonment. The case sparked national outrage and prompted the Dutch Public Prosecution Service to commission the Posthumus Report, which uncovered extensive investigative and prosecutorial failings [2]. As a result, major reforms were introduced, including improved oversight of criminal investigations and the institutionalization of the 'devil's advocate' approach to challenge prevailing theories and prevent confirmation bias in complex cases [2].

## Bias in complex police investigations

When investigating complex criminal cases, police are often confronted with a range of challenges, including limited resources, high investigative pressure, and ambiguous or missing evidence [3,4]. These factors can make it difficult to piece together an accurate picture of what happened, leading investigators to rely on fragmented or unclear evidence [5]. Although one might expect investigators to uphold the principle of 'innocent until proven guilty' in such cases, research indicates that when evidence is ambiguous and stakes are high, decision-making becomes especially prone to error [4]. This is because the intense pressure to resolve serious crimes – often fuelled by public scrutiny and media coverage – can push investigators to try to solve the case quickly by securing a conviction. Based on first pieces of seemingly incriminating evidence, investigators may develop an initial theory of guilt. Under uncertainty and time pressure, this initial theory of guilt may lead to filtering out contradictory evidence [6] or interpreting ambiguous behaviour as confirming their suspicions [7].

This tendency to seek and prioritize evidence that supports the investigator's initial belief about a suspect's guilt is called confirmation bias [8]. In police investigations, confirmation bias can distort both the collection and interpretation of evidence – undermining investigative objectivity and increasing the risk of wrongful convictions [5, 8]. This often occurs when investigators fixate on a single theory or suspect and disregard alternative explanations [6]. Research consistently demonstrates that confirmation bias is prevalent in criminal investigations (e.g., [5, 9]), often resulting in cascading biases that influence prosecutors, judges, and juries, who may rely on selectively gathered evidence [5]. It is therefore crucial to develop effective methods to combat confirmation bias in police investigations.

Efforts to counter confirmation bias in Dutch criminal investigations have included the introduction of a devil's advocate approach ('Tegenspraak') following the miscarriage of justice in the Schiedammer Park Murder case [2]. In this procedure, a criminal investigator who is not part of the investigative team critically evaluates whether the team's investigative hypotheses and decisions are adequately supported by the available evidence, whether alternative lines of inquiry have been identified and

pursued, and whether rejected theories were responsibly excluded. There is evidence to suggest that *Tegenspraak* can promote reflection and critical thinking within investigative teams, but also that effects are subtle and that effectiveness depends heavily on how open team leaders are to critique [10]. These mixed findings underscore the challenge of developing interventions that effectively target confirmation bias.

## Mitigating confirmation bias in police investigations

A recent review of debiasing strategies by Neal et al. [11] showed that simply raising awareness or issuing warnings about confirmation bias is insufficient to reduce confirmation bias in criminal investigations. Instead, they suggest that it is more efficient to consider why a judgement or theory might be wrong – an approach known as *falsification-based reasoning* [12], which emphasizes attempts to reject one's assumptions. In investigative practice, this idea has been incorporated into methods to mitigate bias known as *alternative scenario* or *alternative hypothesis* approach [13, 14, 8, 15, 16, 17]. Apart from an emphasis on falsification, these approaches encourage investigators to consider multiple potential explanations for the available evidence, rather than focusing on a single theory. These approaches are thought to drive investigators to actively seek information that challenges their hypothesis, reducing susceptibility to confirmation bias and promoting a more objective, comprehensive evaluation of evidence [13, 8, 16].

Several studies have systematically examined alternative scenario approaches, advancing their application across various legal decision-making contexts. Research has explored their role in assessing the reliability of victim statements [15], supporting cold case investigations [3], and promoting falsification-based reasoning through structured tools like the Analysis of Competing Hypotheses [18]. Other work has investigated their potential to mitigate the influence of confession evidence [19] and confirmation bias in forensic case assessments [20, 16]. Together, these studies have contributed to our understanding of how alternative scenario methods can be integrated into investigative and judicial settings, though findings regarding their effectiveness have been mixed (for an overview, see [15]).

Some studies suggest that considering competing scenarios can reduce confirmation bias [14, 8], others report null results [15, 18, 19]. While the null effects are surprising given the scenario method is rooted in current scientific practice (e.g., [16]), multiple studies could not properly test the effectiveness of the debiasing intervention due to a lack of induced bias (e.g., [15, 18]). Possible explanations identified in prior research include the use of student samples, small sample sizes with limited statistical power, failed or insufficient manipulations, and low-participant engagement. In addition to these explanations, one potential reason for the null effects is that alternative scenario methods may still inadvertently reinforce a suspect-focused mindset.

In criminal investigations, there is a natural tendency to focus on identifying the perpetrator – especially in high-profile cases where high investigative pressure encourages quick convictions [4]. However, focusing on a suspect early in the process can exacerbate confirmation bias, discouraging investigators from seeking evidence that challenges their assumptions [3, 4, 20] and increasing the risk of wrongful convictions. This suggests that simply contrasting alternative hypotheses may not be sufficient to reduce confirmation bias, but that it is also critical to shift the focus away from individual suspects. In the absence of clear direction, alternative scenario methods may fail to achieve this shift, as the different scenarios investigators develop can still revolve around (the same) suspects. For example, in both [15] and [18], participants were instructed to consider competing scenarios or hypotheses within a case that was anchored to a central figure. In [18], participants' reasoning and outcome measures focused on confirming or exonerating a main suspect's guilt, while in [15], participants evaluated the reliability of a single witness's statement. Although alternative scenarios were introduced, reasoning in both studies remained focused on evaluating specific individuals, rather than integrating the evidence to reconstruct what might have occurred.

Another reason for the mixed results regarding the effectiveness of alternative scenario methods could be an insufficient emphasis on falsification. While falsification is a key component of these methods [3, 17], most studies have only implicitly encouraged participants to engage in it – for example by asking participants to rate how well evidence fits with different hypotheses (26,

20] or evaluate the importance of evidence for different scenarios [18]. Among these studies that indirectly encourage falsification, only Rassin [8] found significant bias reduction, but this effect was not replicated in Niccolson & Rassin [19].

In contrast, a smaller number of studies have attempted to incorporate falsification more explicitly in their instructions. Maegherman et al. [18] asked participants to decide on a defendant's guilt in a murder case and compared instructions to justify, explicate, or falsify one's decision. While no differences were observed in guilt perception, falsification and explication instructions led to greater consideration of exonerating evidence, which predicted acquittal rates. O'Brien [14] found that explicitly instructing participants to falsify their initial hypothesis led to reduced confirmation bias, whereas merely generate alternative hypotheses did not.

This suggests that while generating alternative hypotheses may broaden perspective, falsification requires a mindset shift: from generating options, to actively attempting to refute one's own favoured theory. The review by Neal et al. [11] supports this, showing that debiasing strategies are most effective when they prompt individuals to consider why a hypothesis might be incorrect. As Rassin [8] and Niccolson and Rassin [19] argue, investigators may not naturally reject their hypotheses based solely on imperfect evidence or conflicting information, underscoring the need for explicit guidance to focus on disconfirming evidence.

### The scenario reconstruction method

The Scenario Reconstruction Method is an alternative scenario approach developed for the Dutch police following the Schiedammer Park Murder case, in response to the Posthumus Report [2] that called for tools to counter confirmation bias in investigations [3]. It is built around two core components: 1) shifting the investigative focus from individual suspects to the available evidence, and 2) actively promoting falsification. Investigators reconstruct multiple scenarios based on the available evidence, concentrating on the question 'What happened?' rather than 'Who did it?' [3]. They then attempt to falsify each scenario by identifying pieces of evidence that can discriminate between them. This approach helps investigators remain open to alternative explanations, critically evaluate evidence, and identify key investigative steps needed to rule out specific scenarios [21].

Building on this framework, our study uses these core elements of the Scenario Reconstruction Method to test whether 1) shifting focus from suspects to evidence and 2) explicitly encouraging falsification can reduce confirmation bias in police investigations.

### The current study

This study examines whether promoting an investigative focus on evidence rather than suspects, and on falsification rather than verification, can reduce confirmation bias and improve investigative accuracy. Specifically, we independently manipulated both the *investigative focus* (suspect- vs. evidence-focused) and the *reasoning strategy* (verification- vs. falsification-focused) to examine how each shapes investigative reasoning. Current and future police officers read a criminal case, based on the Schiedammer Park Murder, that initially biases toward an innocent suspect – 'Kees B'. They analyse the case, giving their initial assessment, with a focus on either the evidence (what happened?) or the suspect (who did it?). Then, they receive new evidence, including exonerating information on Kees B, and evaluate which pieces of evidence prove them right (verification) vs. wrong (falsification). We hypothesize that strategies emphasizing a focus on evidence (vs. suspects) and falsification (vs. verification) results in more accurate (i.e., lower) guilt ratings for the prime suspect, Kees B. As a secondary outcome, we expected more evidence- and falsification-focused suggested next investigative steps, in the evidence- and falsification-focused conditions, compared to the suspect- and verification-focused conditions, respectively.

## Materials and methods

The materials, coding scheme, data, as well as all supplementary appendices are available at: https://doi.org/10.6084/m9.figshare.30518270.

## Ethics

This study was approved by the Ethics Review Board of the Faculty of Social and Behavioral Sciences, University of Amsterdam, The Netherlands, and archived under number FMG-8701/FMG-8702. Participants provided digital informed consent.

## Sample

An a priori frequentist power analysis conducted with G*Power [22] indicated a required sample size of $N = 231$ for detecting moderate effects (Cohen's $d = 0.50$) with a power of 0.90 and an alpha level of.05, in the primary two-way ANOVA. Notably, all our analyses were Bayesian, which typically yield robust inferences with smaller samples, but the power analysis provided a conservative benchmark. Moreover, a related study [14] reported a larger effect size of approximately $d = 0.60$. To account for an estimated dropout rate of 10%, we aimed for a final sample size of $N = 260$.

Participants were recruited between January 20, 2025 and February 14, 2025 through two contacts at the German Police Academy in Villingen-Schwenningen, as well as through personal connections within the German police. To incentivize participation, participants had the chance to win a voucher for an online shop of their choice (e.g., Amazon, Spotify, or Zalando). One €20 voucher was raffled for every 50 participants, and one €100 voucher for every 100 participants.

441 people started the study, but 147 did not complete it and were excluded. Overall, attention check performance was extremely high (95.6% correct responses for attention check 1, 99.6% for attention check 2 & 3, and 98.6% correct responses for attention check 4), indicating that the participants understood and actively engaged with the materials. We excluded the one participant who answered more than one attention check incorrectly. Manipulation check performance was also high, with 96.6% of participants passing the focus manipulation check and 84.0% passing the strategy manipulation check, indicating that most participants adhered to the intended focus and applied the intended reasoning strategies. No participants were excluded based on manipulation check performance, but we took the manipulation checks into account in a per-protocol analysis (see below). Thus, the final sample consisted of $N = 293$ participants with a mean age of 25.13 years ($SD = 6.67$) and average practical experience with police work of 3.10 years ($SD = 5.83$; range 0–42 years). The gender distribution was approximately equal (48.5% male, 51.5% female). About half (56.0%) reported current or previous affiliation with the criminal investigation office, the division within the German police which handles serious crimes, such as homicide or sexual offenses. Most participants (71.3%) were currently undergoing initial (~ 52.2%) or continuing training (19.1% – participants who continued their training previously worked as patrol officers). Participants were primarily recruited through instructors at the police academy (78.5%), with the remainder recruited via other channels (11.6%), their superiors (5.1%), colleagues (3.8%), or friends (1%).

## Procedure

The study was conducted online and employed a 2 × 2 between-subjects design, manipulating the variables focus (suspect vs. evidence) and strategy (verification vs. falsification). Participants were randomly assigned to one of the four experimental conditions, (Fig 1). These two manipulations targeted independent aspects of investigative reasoning. The focus manipulation directed *what* participants' reasoning was organized around (a suspect vs. the evidential reconstruction of events). The reasoning-strategy manipulation directed *how* they approached their current assumptions (verification vs. falsification).

After providing digital informed consent, they read a vignette about the Schiedammer Park Murder including information about the prime suspect Kees B (renamed 'Robert' in the actual study materials, as to avoid recognition of the case) and the 12-year-old victim M, who also came under suspicion. Their order of introduction was randomized. After reading the case vignette participants completed three attention checks. Depending on the condition, participants either identified the suspect they believed committed the crime (suspect-focus) or outlined two plausible scenarios for what might have

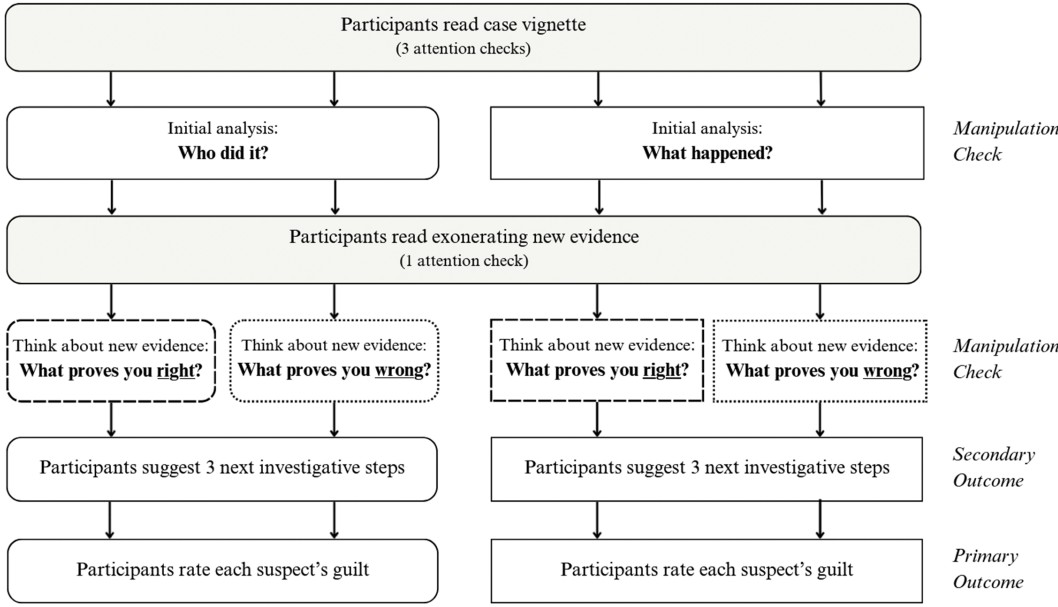

**Fig 1. Overview of experimental design.** This figure shows the 2 × 2 experimental design and task sequence.

happened (evidence-focus). They were then presented with new evidence, including exonerating information about Kees B that specified the timeline of events and indicated that he could not have reached the park in time to commit the crime. Afterward, they completed another attention check. In line with their assigned strategy, participants then listed how the new evidence either supported (verification focus) or contradicted (falsification focus) their prior response.

Next, participants proposed three investigative steps that should be taken next in the investigation, explaining what actions should be taken and why (our secondary outcome measure). They also rated the perceived guilt of the prime suspect Kees B, the 12-year-old victim M, and an unknown perpetrator – on a scale from 0 to 100 (our primary outcome measure). They provided a justification for their judgment in a free-text box (this was an exploratory measure and not analysed further). The survey concluded with demographic questions, including age, gender, level of police training, experience in practical police work, and affiliation with the criminal investigative office.

## Materials

The complete set of study materials (original: German; translated to English) is provided in S2 Appendix. Below we summarize its key elements, in the order in which participants processed them.

**Case vignette.** The case vignette is based on a well-documented case of wrongful conviction, the Schiedammer Park Murder [1]. We based our case vignette on this real-life example because it clearly illustrates how confirmation bias can derail a criminal investigation. This makes it a suitable context for testing strategies intended to reduce such bias. The vignette is structured to initially elicit a bias toward the primary suspect, 'Kees B'. In the vignette, Kees B appears highly suspicious of having attacked the two children – 9-year-old Nina (who died) and 12-year-old M (who survived by pretending to be dead) – because weeks before, in the same park, he had offered a boy money in return for sex, and he confessed to committing the murder and entering the park that day with the intention of seeking sexual contact with children. We conducted a pilot study ($n = 36$) on Prolific to examine whether this initial case information induced the intended suspicion towards Kees B. The pilot participants only read the initial case information and – without completing the remainder of the procedure – immediately provided their guilt ratings. The guilt ratings for Kees B ($M = 51.05$,

$SD = 25.42$) was found to be higher than that of M ($M = 23.76$, $SD = 18.85$) $d = 0.78$, $BF_{+0} = 2923.78$, however not higher than someone else's guilt ($M = 41.34$, $SD = 29.70$) $d = 0.21$, $BF_{0+} = 1.65$. Due to time constraints, we had to proceed with the study regardless, but the most important outcome of the pilot was that Kees B was clearly perceived as more likely guilty than M based on the initial information.

**Focus: Evidence versus suspect.** We asked participants to indicate who did it (suspect-focus) or what happened (evidence-focus), based on the evidence from the case vignette. Specifically, in the suspect-focused conditions, participants were asked to identify who they think is the most likely suspect and list evidence incriminating this suspect. In the evidence-focus conditions, participants were asked to reconstruct two possible scenarios based on the available evidence. These responses were later evaluated as the focus manipulation check using the coding criteria outlined in S1 Appendix.

**Exonerating evidence.** The new evidence was designed so that it was exonerating but leaves some room for interpretation of Kees B's guilt. For instance, while the initial evidence established that the crime occurred between 17:15 and 17:34, the new evidence reveals that Kees B worked 11 minutes away from the park and, based on his clock-out time, could not have arrived earlier than 17:27. This would have given him a maximum window of 7 minutes to find the children, lead them into the bushes, commit the crime, and flee. This makes his involvement highly improbable, though not entirely impossible.

**Strategy: Falsification versus verification.** We instructed participants to list ways in which the exonerating evidence supports (verification) or contradicts (falsification) their initial analysis. These responses were evaluated as the strategy manipulation check using the coding criteria detailed in S1 Appendix.

**Next investigative steps.** By deciding what steps should be taken in an investigation, investigators shape the information that ultimately informs judicial decisions about a suspect's guilt or innocence. Therefore, promoting objectivity throughout the investigative process is critical to ensure accurate and fair outcomes. Encouraging strategies that emphasize evidence-based reasoning and the active testing of assumptions through falsification may increase the reliability of investigative findings. To examine how the different investigative strategies influenced participants' approach to advancing the case, they were asked to propose three next steps they believed should be taken in the investigation. For each step, participants were asked to describe what should be done and explain why. Each proposed step was then evaluated on two dimensions: whether it reflected a suspect- or evidence-focused approach, and whether it aimed at verification or falsification/critical testing of prior assumptions.

**Guilt rating.** Participants were asked to indicate how certain they are that Kees B, M or someone else is guilty on a scale from 0 (not guilty) to 100 (certainly guilty). Since Kees B is initially presented as the most likely perpetrator but is in fact innocent, lower guilt ratings for him are considered more accurate. In this study, accuracy refers to how closely participants' conclusions matched the material truth, meaning the factual sequence of events and offender identity as established in the case records. This may differ from the legal truth, which reflects the interpretation reached through judicial proceedings.

**Attention checks.** Four attention checks were embedded in the study to ensure that participants understood and engaged with the materials (Fig 1). Three were presented after the case vignette and one after the exonerating evidence. Each attention check consisted of a single multiple-choice question with four response options, only one of which was correct. For example: 'How did the children initially notice the time?' with the options: a) They checked their watches, b) They asked a passerby, c) They saw it on a nearby clock, d) It didn't say.

## Coding

The full coding scheme can be found in S1 Appendix.

**Coding of focus manipulation check.** Participants' responses were rated on a 5-point scale ranging from –2 (fully suspect-focused) to +2 (fully evidence-focused), reflecting the extent to which participants followed the assigned

instructions: either to identify a suspect and list incriminating evidence (suspect-focus) or to reconstruct two possible scenarios based on the available evidence (evidence-focus). For example, a response such as 'I think Kees B did it because he has paedophilic interests and confessed to the murder' would be coded as –2 (fully suspect-focused), as it identifies a suspect and lists incriminating evidence. Another example is '*Re-interview M to see if his story about where they played matches the new witness statement*', which would be coded as mostly evidence-focused (as the primary purpose is to clarify what happened, rather than to assess M's guilt. Responses are coded as 0 (mixed) if they contain both suspect- and evidence-focused elements. They are coded as NA if the focus is unclear or cannot be clearly assigned to either (sub-)category. Per participant, we consider the focus manipulation to have failed if 1) the response is coded as 0 (mixed) or NA (uncodable), or 2) if participants in the suspect-focused condition provide an evidence-focused response, and vice versa.

**Coding of strategy manipulation check.** Participants' responses were rated on a 5-point scale ranging from –2 (fully verification-focused) to +2 (fully falsification-focused), based on the extent to which they complied with the instruction to either list points that support or contradict the participant's initial assessment. For instance, the response '[I think Kees B is the perpetrator] – His bike was seen near the crime scene, and he worked nearby so he could have made it to the park in time to commit the crime' would be coded as –2 (fully verification-focused), as it introduces information that supports the original conclusion. Responses are coded as 0 (mixed) if they contain both verification- and falsification-focused elements (e.g., verifying one scenario while falsifying another). Responses are coded as NA if it is unclear—based on the answers to both manipulation checks—whether the response is meant to verify or falsify a prior assumption. For example, if the focus manipulation check identifies two scenarios (e.g., 1) M did it, 2) Kees B did it), but the strategy manipulation check response lists 'time of crime, confession' without specifying how these points relate to either scenario, it is not possible to determine whether the response is aimed at verifying or falsifying either scenario. Per participant, the strategy manipulation is considered failed if 1) the response is coded as 0 (mixed) or NA (uncodable), or 2) if participants in the verification-focused condition provide a falsification-focused response, and vice versa.

**Coding of next investigative steps.** Each proposed investigative step was evaluated along two dimensions. First, the step's suspect- vs. evidence-focus was rated on a 5-point scale (–2 to +2), reflecting the extent to which it centred on a particular suspect versus the broader evidentiary context. Second, the step's verification- vs. falsification-focus was rated on a 3-point scale (–1 to +1), based on whether the step aimed to confirm or critically test existing assumptions. For example, the step 'Question Kees B's colleagues to see if he left work straight after clocking out – if he didn't, he couldn't have made it to the park in time' would be coded as suspect-focused (as it pertains directly to Kees B as a potential suspect) and falsification-focused (as it critically tests whether he could have been involved). Responses are coded as NA if it is unclear whether they 1) focus on suspects or evidence, and 2) aim to verify or falsify a prior assumption.

**Evaluation of coding quality.** The coding was practiced and refined on pilot data ($n = 57$). One condition-blind annotator coded all the responses, and one condition- and hypothesis-blind coder coded a random 20% of the responses as to estimate inter-rater reliability.

For the focus manipulation check, inter-rater agreement was moderate to substantial, with percent agreement = 0.71, Cohen's Kappa = 0.60, and PABAK = 0.65. This suggests a high level of consistency between raters. 10 responses were classified as manipulation failures.

For the strategy manipulation check, inter-rater agreement was fair to moderate, with percent agreement = 0.51, Cohen's Kappa = 0.34, and PABAK = 0.41. This reflects moderate challenges in applying the coding scheme consistently across raters. 47 responses were considered manipulation failures.

For the next investigative steps, inter-rater agreement was fair to moderate on both dimensions. For suspect- vs. evidence-focus, percent agreement = 0.51, Cohen's Kappa = 0.34, and PABAK = 0.41. For verification- vs. falsification-focus, percent agreement = 0.56, Cohen's Kappa = 0.36, and PABAK = 0.41. These results indicate some inconsistencies between coders, suggesting that this aspect of the coding scheme may have been more subjective or difficult to apply reliably.

 

## Results

### Primary outcome: Guilt ratings

We expected that evidence- and falsification-focused investigative strategies – individually and in combination – would lead to more accurate guilt ratings (i.e., lower ratings for Kees B) compared to suspect- and verification-focused strategies. To evaluate this, we conducted a Bayesian ANOVA using JASP (version 0.19.1, [23]), with two between-subjects factors: focus (suspect-focused vs. evidence-focused) and strategy (verification vs. falsification) on Kees B's guilt rating. All analyses used the default Cauchy-prior with distribution centred at zero. The analysis compared five models: the null model, each main effect separately, the additive model with both main effects, and the full model including their interaction.

Table 1 presents the Bayes factors and posterior model probabilities for all models. The null model, which includes only an intercept, received the strongest support (posterior probability = 0.752). All models including main or interaction effects were substantially less supported. The best of these (focus only) had a posterior probability of 0.118 and a Bayes factor of $BF_{10} = 0.157$, indicating the data were approximately 6.4 times more likely under the null than under a model including only the focus factor. These results suggest that none of the experimental manipulations substantially improved the model's predictive performance relative to the null.

To assess the evidence for including each predictor across the entire model space, we examined inclusion Bayes factors. Evidence was weak for both main effects ($BF_{incl} = 0.106$ for focus; $BF_{incl} = 0.100$ for strategy), and virtually non-existent for the interaction ($BF_{incl} = 0.012$). These values indicate strong evidence against including any of the predictors, consistent with the model comparison.

Despite the lack of model support, we examined the model-averaged posterior estimates for each factor level (Table 2). All credible intervals for the condition-level effects included zero, reflecting the lack of evidence for systematic differences in guilt ratings across groups.

In sum, the Bayesian ANOVA revealed strong evidence in favour of the null model, suggesting that neither focusing on evidence (vs. suspects) nor falsifying (vs. verifying) the initial analysis, nor their interaction, reliably influenced Kees B's guilt rating. Inclusion Bayes factors and posterior distributions further corroborated the absence of effects, with all 95% credible intervals overlapping zero. For reference, Table 3 shows Kees B's average guilt across conditions.

### Secondary outcome: Next investigative steps

We expected 1) that participants instructed to focus on evidence would propose more evidence-focused next investigative steps than participants who were instructed to focus on identifying a suspect, and 2) that participants who were told to falsify their initial analysis would propose more falsification-focused next steps than participants who were told to verify their initial analysis. We examined this by fitting two separate Bayesian ordinal mixed-effects models in R [24], one examining

**Table 1. Bayesian ANOVA model comparison for guilt ratings.**

| Model | P(M) | P(M\|data) | $BF_{10}$ |
|---|---|---|---|
| Null model | .200 | **.752** | 1.000 |
| Focus | .200 | .118 | 0.157 |
| Strategy | .200 | .111 | 0.148 |
| Focus + Strategy | .200 | .016 | 0.021 |
| Focus + Strategy + Focus * Strategy | .200 | .003 | 0.004 |

*Note.* P(M) is the prior model probability; P(M|D) is the posterior model probability given the data; $BF_{10}$ is the Bayes factor relative to the best model (here, the null). Bolded value indicates the model with the highest posterior probability.

**Table 2. Model-averaged posterior summaries for guilt ratings.**

| Variable | Level | Mean | SD | 95% CI Lower | 95% CI Upper |
|---|---|---|---|---|---|
| Intercept | — | 43.616 | 1.626 | 40.307 | 46.808 |
| Focus | Suspect | −0.768 | 1.301 | −3.381 | 1.828 |
|  | Evidence | 0.768 | 1.301 | −1.855 | 3.354 |
| Strategy | Falsification | 0.666 | 1.281 | −1.899 | 3.214 |
|  | Verification | −0.666 | 1.281 | −3.239 | 1.874 |
| Focus × Strategy | Suspect & Falsification | 0.369 | 1.271 | −2.173 | 2.896 |
|  | Suspect & Verification | −0.369 | 1.271 | −2.919 | 2.149 |
|  | Evidence & Falsification | −0.369 | 1.271 | −2.919 | 2.149 |
|  | Evidence & Verification | 0.369 | 1.271 | −2.173 | 2.896 |

**Table 3. Mean guilt ratings for Kees B across conditions.**

| Kees B | Suspect | | Evidence | |
|---|---|---|---|---|
|  | Verification | Falsification | Verification | Falsification |
| **Mean** | 41.98 | 43.97 | 44.32 | 44.70 |
| **SD** | 21.28 | 22.82 | 22.95 | 23.37 |

*Note.* Lower guilt ratings reflect higher accuracy.

the effect of focus (suspect vs. evidence), the other the effect of strategy (verification vs. falsification) on the type of next investigative steps participants suggested. The first model showed that participants instructed to focus on evidence were indeed more likely to propose evidence-focused next steps than those instructed to focus on suspects, $\beta = 0.48$, SE = 0.19, 95% CI [0.10, 0.86]. Similarly, the second model indicated that participants told to falsify their initial analysis were more likely to suggest next steps aimed at falsifying or critically testing an assumption than those told to verify their initial analysis, $\beta = 0.51$, $SE = 0.21$, 95% CI [0.09, 0.93]. Both effects were statistically credible, as the 95% credible intervals excluded zero. Fig 2 shows the suspect- vs. evidence- and Fig 3 the verification- vs. falsification-scores of the suggested next steps across conditions.

### Exploratory analyses

**Interaction effects and cross-over patterns of focus and strategy on next investigative steps.** As an exploratory extension of the secondary analyses, we examined whether focus (suspect vs. evidence) and strategy (verification vs. falsification) interacted in shaping participants' suggested next investigative steps. The interaction effects on both outcomes were inconclusive. For the suspect- vs. evidence-focus of the suggested next steps, the interaction was $\beta = -0.51$, $SE = 0.27$, 95% CI [−1.06, 0.01]; for their verification- vs. falsification-focus, $\beta = -0.57$, $SE = 0.31$, 95% CI [−1.18, 0.03]. In both cases, the 95% credible intervals included zero, indicating that the data do not provide strong evidence for an interaction effect, and that the true effect size could plausibly be close to zero.

Interestingly, we observed cross-over effects: 1) participants instructed to focus on evidence were more likely to propose next steps that aimed to falsify or critically test an assumption ($\beta = 0.40$, $SE = 0.19$, 95% CI [0.04, 0.76]) than participants instructed to focus on suspects, and 2) participants who were instructed to falsify their initial analysis were more likely to suggest next investigative steps centred around collecting or evaluating evidence ($\beta = 0.61$, $SE = 0.22$, 95% CI [0.18, 1.06]) than participants instructed to verify their initial analysis. The 95% credible intervals for both cross-over

**Fig 2. Mean suspect- vs. evidence-focus of suggested next steps across conditions.** Average suspect and evidence strategy ratings of the next investigative steps generated in each experimental condition. Scores below 0 indicate suspect focused reasoning, and scores above 0 indicate evidence focused reasoning. Error bars represent ±1 SE.

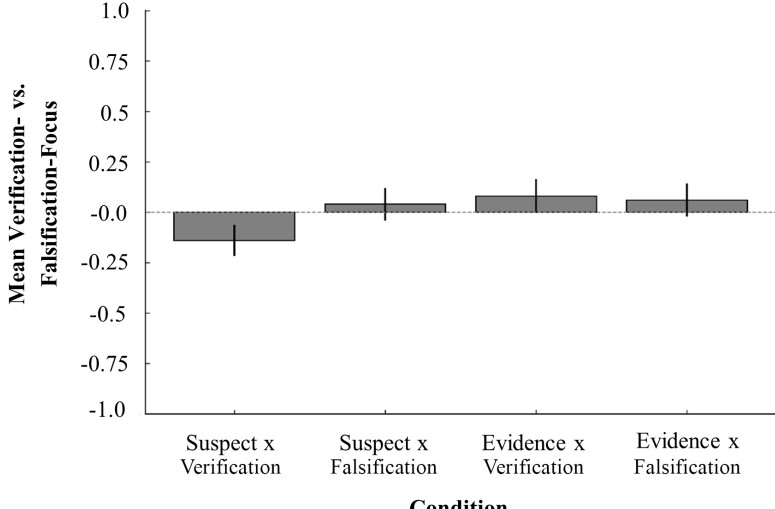

**Fig 3. Mean verification- vs. falsification-focus of suggested next steps across conditions.** Average verification and falsification focus ratings of the next investigative steps generated in each experimental condition. Scores below 0 indicate verification focused reasoning, and scores above 0 indicate falsification focused reasoning. Error bars represent ±1 SE.

effects excluded zero, suggesting that each manipulation may have influenced participants' reasoning across both dimensions.

**Effectiveness under ideal versus real-world conditions.** To examine how effective the investigative strategies were under both 'ideal' and 'real-world' conditions, we followed an approach similar to a per-protocol analysis (see [25]), which includes only participants who followed the experimental instructions as intended and thus allows us to assess the strategies' effects under optimal conditions. Specifically, we repeated both the primary analysis (evaluating the impact of

the different investigative strategies on guilt judgment accuracy) and the secondary analyses (evaluating the influence of the different investigative strategies on the type of suggested next investigative steps), using only participants who passed both manipulation checks ($n = 237$).

The exclusion of participants who failed either manipulation check had no discernible impact on the accuracy of guilt ratings. Bayes factors, effect sizes, and credible intervals remained highly similar to those observed in the full-sample analysis. For the effects on the suspect- vs. evidence-focus of the next investigative steps, excluding participants who failed either manipulation check weakened the effects, so that the previously observed main- and cross-over effects were smaller and no longer credible. Similar results were observed for the verification- vs. falsification-focus of the next investigative steps – here, too, the main effect of falsification-focused strategies disappeared when including only participants who passed both manipulation checks. However, the cross-over effects of falsification-focus on evidence-focused strategies remained.

**Influence of covariates on accuracy of guilt ratings and types of suggested next steps.** We conducted an exploratory Bayesian ANCOVA in JASP [23] to examine whether the accuracy of Kees B's guilt rating was influenced by individual differences, including age, gender, practical policing experience, level of police training, and affiliation with the criminal investigation office. The independent variables – focus (suspect vs. evidence) and strategy (verification vs. falsification) – were included as fixed factors, while the aforementioned individual differences were entered as covariates. Among these covariates, affiliation with the criminal investigation office was the only variable for which the data provided strong evidence of an effect. Participants who were affiliated with the criminal investigation office tended to assign higher guilt ratings to the innocent suspect Kees B, $\beta = 9.11$, $SD = 2.60$, 95% CI [3.90, 14.25], $BF_{incl} = 94.13$, $P(incl|data) = .958$. For all other covariates, the evidence favoured the null model (all $BF_{incl} < 0.41$), suggesting that these variables were not credibly associated with the accuracy of guilt ratings.

## Discussion

High-profile miscarriages of justice – such as the Schiedammer Park Murder case in the Netherlands [1] and numerous wrongful convictions uncovered by the Innocence Project [26] – have spurred major reforms in investigative procedures. Among these reforms, the Scenario Reconstruction Method [3] has been introduced as a promising tool to reduce confirmation bias in police investigations. While its comprehensive, theory-driven approach is gaining institutional support, we still know little about its effectiveness, or which specific components are responsible for its potential debiasing effects.

This study addresses that gap by applying a dismantling approach to isolate and evaluate two key elements of the method: 1) shifting the investigative focus from suspects to reconstructing possible scenarios based on available evidence, and 2) encouraging officers to falsify rather than verify their assumptions. Drawing on these principles from the Scenario Reconstruction Method [3], we employed a 2×2 design with current and future police officers from Germany to test how focus (suspect vs. evidence) and reasoning strategy (verification vs. falsification) influence guilt judgment accuracy and the type of next investigative steps participants suggest. In doing so, this study offers a more fine-grained and ecologically valid test of debiasing strategies, contributing to both theoretical development and practical improvements in investigative decision-making.

### Manipulating focus and strategy did not influence accuracy of guilt ratings

We expected that encouraging participants to focus on the available evidence rather than a specific suspect, to try to falsify rather than confirm their beliefs, and especially to do both, would lead to more accurate ratings of an innocent suspect's guilt. However, these expectations were not supported: none of the strategies – on their own or combined – led participants to judge the suspect's guilt more accurately. Given the high attention check performance (over 95%), strong manipulation check pass rates (96.6% and 84.0%), our large sample ($N = 293$), and the use of Bayesian analyses, this lack of effect is unlikely due to lack of power or participant inattention. So, what could explain it?

One possible explanation for why the interventions did not exert their expected influence, is that they may have been too brief to meaningfully disrupt entrenched cognitive biases like confirmation bias [9, 11, 27, 28]. Prior research suggests that debiasing interventions are more likely to be effective when implemented as part of a sustained, structured process – especially in complex decision-making contexts where individuals must interpret ambiguous information and operate with a high degree of autonomy [29, 27, 11, 28]. Brief, cognitively demanding instructions, on the other hand, may inadvertently increase mental load and backfire by making participants to fall back on intuitive reasoning, particularly when under time pressure or uncertainty [27, 28].

Given these challenges, it is increasingly recognized that successful debiasing likely requires systematic support across multiple stages of the decision-making process, rather than isolated or one-time interventions. In forensic psychological practice, bias-mitigating strategies such as transparent documentation, the generation of alternative hypotheses, and critical peer challenge are not employed in isolation but are integrated consistently throughout the assessment process to guide evaluators at every step [28]. Similarly, the Scenario Reconstruction Method is designed as a comprehensive, staged procedure intended to support investigative reasoning continuously [3]. It is important to note that, the present findings do not reflect the effectiveness of the full method, as we tested only two isolated aspects: shifting the focus from suspects to evidence and encouraging the falsification of assumptions. While examining the contribution of individual components remains valuable to evaluate key working mechanisms, it is plausible that the synergistic application of multiple components is necessary to achieve reliable bias mitigation. Therefore, future research should evaluate the Scenario Reconstruction Method holistically, ideally through randomized controlled trials that assess its effectiveness as a fully integrated procedural approach to reducing confirmation bias [21].

## Promoting a focus on evidence and falsification improves proposed investigative strategy

Participants who were encouraged to engage with the case materials by focusing on the available evidence – rather than on identifying a likely suspect – subsequently proposed investigative steps that prioritized collecting or evaluating evidence independent of specific individuals. In contrast, those instructed to consider who might be the most likely suspect were more inclined to suggest steps that centred around investigating individual suspects. Similarly, participants who were told to use new evidence to falsify their prior analysis later proposed steps aimed at critically testing assumptions or distinguishing between competing scenarios by seeking disconfirming evidence. Interestingly, those instructed to focus on evidence were also more likely to suggest falsification-focused steps, and vice versa. This pattern suggests that the manipulation may have shaped participants' broader investigative mindset, promoting critical thinking that extends beyond immediate, instruction-specific responses.

That participants' reasoning shifted is encouraging given that many debiasing strategies – even when conceptually sound – often fail to produce changes in final evaluative judgements, such as assessments of suspect guilt or statement reliability (e.g., [30, 15, 18, 19]. The present findings suggest that while final judgements in investigative processes may be relatively resistant to change, early-stage investigative processes may be more receptive to debiasing efforts, Figs 2 and 3. This aligns with theoretical perspectives emphasizing that debiasing interventions are more likely to succeed when applied during earlier phases of decision-making, when uncertainty remains high and individuals are still open to critical reflection and careful evaluation [29, 11, 27]. These findings underscore the importance of developing strategies that intervene early in the investigative process, before individuals commit to a particular interpretation or conclusion, thereby enhancing cognitive flexibility and promoting the objective consideration of multiple hypotheses.

The observed cross-over effects further support this interpretation. Rather than simply prompting participants to follow the instructed strategies, the interventions appear to have fostered a broader investigative mindset characterized by critical thinking and openness to alternative explanations. For example, encouraging participants to focus on the available evidence – rather than identifying suspects – may foster critical thinking, as the evidence often does not clearly point in one direction and may thereby prompt more questioning and reflection (as indicated by the more falsification-focused

suggested next steps). In contrast, forming an early theory about a suspect naturally seems more likely to facilitate a selective interpretation of incoming evidence. Cultivating critical and open mindsets early in the investigative process may thus be key to reducing biases [13, 21, 17]. An example of how to concretely promote this mindset is the Scenario Reconstruction Method, or SRM [3]. In this approach, investigators first focus on reconstructing what happened before evaluating who might be responsible. Individuals are initially treated as persons of interest. That is, as individuals relevant to the event but not yet linked to any specific hypothesis of guilt. This distinction ensures that suspect hypotheses follow the evidential reconstruction rather than preceding it, helping prevent premature assumptions of culpability (see [21], for a practical overview of SRM in Dutch, or [3] for an overview in English).

Taken together, the findings suggest that even small, targeted interventions can meaningfully shape the investigative process when applied before strong commitments form. As this may offer a promising avenue for fostering more objective and reliable investigative outcomes, future research should aim to replicate these effects and determine whether and how early-stage investigative reasoning can be reliably influenced across different samples and contexts. One key question is whether such strategies are more effective when delivered before exposure to case materials and the formation of an initial interpretation. Presenting debiasing instructions at this early stage – such as encouraging evidence-focused reasoning and the active falsification of assumptions – may help prevent biased reasoning from arising in the first place, rather than attempting to correct it after it has already taken hold [28].

It is also theoretically plausible that proposing more objective investigative steps could yield more accurate guilt ratings over time, particularly if investigators are given feedback on the outcomes of their suggested steps [31]. In the present study, participants did not receive feedback on the investigative steps they suggested, so they could not observe whether these steps would have uncovered exonerating evidence or corrected initial impressions. As a result, we cannot determine whether the more objective investigative strategies promoted by the interventions would have ultimately led to better outcomes. Future studies should therefore simulate investigative feedback, allowing participants to experience the consequences of their information-gathering choices and evaluating whether early improvements in reasoning translate into more accurate investigative outcomes over time.

## Limitations

First, one may argue that there was a degree of overlap between the manipulations and the dependent variables. Specifically, one might argue that participants instructed to adopt an evidence- or falsification-focus simply repeated these instructions when proposing investigative steps, without undergoing a cognitive shift. However, several findings suggest that superficial compliance is unlikely to account for the observed effects. Most notably, we observed cross-over effects – participants in evidence-focus conditions more frequently proposed falsification-focused steps, and vice versa – despite these patterns not being directly prompted by the instructions. This supports the interpretation that the manipulations influenced participants' broader investigative mindset.

Second, the interrater reliability of the coding was mostly fair to moderate. Some responses were difficult to categorize, for instance when participants reasoned that the new evidence verified one scenario but falsified another. We coded such mixed responses as NA. Future studies could streamline this by simplifying both response generation and coding.

Third, it remains possible that the initial bias induction did not produce a strong bias toward Kees B as intended. Our pilot study showed that the case materials successfully biased participants toward viewing Kees B as the most likely perpetrator. However, in the main study, we deliberately avoided asking who participants initially believed was the culprit, to prevent reinforcing a suspect-focus – especially in the evidence-focused conditions. As a result, we lacked a direct measure of whether the bias induction was effective. Post-hoc analyses of the focus manipulation check (who did it/what happened?) suggest some uncertainty: among participants whose responses clearly identified a most likely suspect ($n = 159$), more named one of the victims, 12-year-old M, than Kees B. Although this is an imperfect indicator, it raises the possibility that the bias induction may not have succeeded. Since debiasing strategies are most effective when a strong pre-existing

bias is present [28], the absence of such a bias may have limited the interventions' impact. Future studies should include a condition or measure to verify whether the intended bias has taken hold before applying corrective strategies.

## Practical implications

**Police experience and bias.** The exploratory finding that affiliation with the criminal investigation office was associated with more inaccurate guilt ratings highlights the need to address how professional context may influence evaluative reasoning. Participants who, based on their routine work, can be expected to have greater expertise in handling serious crimes – like the case from this study – unexpectedly assigned higher guilt ratings to an innocent suspect than less specialised officers. This suggests that institutional norms, role-specific expectations, or repeated exposure to high-stakes investigations may reinforce biased or unobjective reasoning patterns [6]. These findings underscore that professional experience alone does not protect investigators against cognitive bias and may even exacerbate it [32, 33]. Embedding debiasing strategies like the Scenario Reconstruction Method into routine training, supervision, and case review procedures may therefore be critical – not only for trainees, but also for experienced investigators – to promote more objective investigative reasoning and reduce the risk of confirmation bias across all levels of investigative practice and various investigative contexts [21, 3].

## Conclusion

This study contributes to advancing the understanding of investigative bias by using a well-powered and ecologically valid sample of police professionals. Brief manipulations encouraging evidence-focused and falsification-based reasoning did not directly mitigate bias. The strategies did reliably shift participants' proposed investigative steps toward greater objectivity, suggesting that such interventions can influence early-stage reasoning even if they do not immediately alter evaluative outcomes. Future work should assess whether early cognitive shifts lead to downstream improvements and whether comprehensive interventions like the Scenario Reconstruction Method yield stronger effects on bias reduction in policing.

## Supporting information

**S1 Appendix. Coding scheme.** Instructions and coding scheme for evaluating the open-text responses to the focus manipulation check, the strategy manipulation check, and the suggested next investigative steps.
(DOCX)

**S2 Appendix. Study materials.** Full case vignette, manipulation texts used across conditions, and demographic questions.
(DOCX)

## Acknowledgments

We are very grateful to Dr. Joachim Albrecht and Dr. Georg Laub from the Hochschule für Polizei Baden-Württemberg for their generous support with data collection. Their help was truly above and beyond, and the strength of our sample would not have been possible without their involvement.

## Author contributions

**Conceptualization:** Sarah Lenz, Tara Zohrevand, Eric Rassin, Bruno Verschuere.

**Data curation:** Sarah Lenz.

**Formal analysis:** Sarah Lenz.

**Investigation:** Sarah Lenz, Tara Zohrevand.

**Methodology:** Sarah Lenz, Tara Zohrevand, Eric Rassin.

**Project administration:** Bruno Verschuere.

**Software:** Sarah Lenz.

**Supervision:** Bruno Verschuere.

**Visualization:** Sarah Lenz.

**Writing – original draft:** Sarah Lenz.

**Writing – review & editing:** Tara Zohrevand, Eric Rassin, Bruno Verschuere.

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
