## [Decision Letter · Decision Letter 0]

18 Jul 2025

PONE-D-25-29848What Happened and What Proves You Wrong? Combating Confirmation Bias in Police Investigations Through Evidence Reconstruction and FalsificationPLOS ONE?

Dear Dr. Verschuere,

Thank you for submitting your manuscript to PLOS ONE. After careful consideration, we feel that it has merit but does not fully meet PLOS ONE’s publication criteria as it currently stands. Therefore, we invite you to submit a revised version of the manuscript that addresses the points raised during the review process.

We look forward to receiving your revised manuscript.

Kind regards,

Bartosz Wojciech Wojciechowski, Ph.D.

Academic Editor

PLOS ONE

Additional Editor Comments (if provided):

Reviewers' comments:

Reviewer's Responses to Questions

**Comments to the Author**

1. Is the manuscript technically sound, and do the data support the conclusions?

Reviewer #1: Yes

Reviewer #2: Partly

2. Has the statistical analysis been performed appropriately and rigorously?

Reviewer #1: Yes

Reviewer #2: Yes

3. Have the authors made all data underlying the findings in their manuscript fully available?

Reviewer #1: Yes

Reviewer #2: Yes

4. Is the manuscript presented in an intelligible fashion and written in standard English?

Reviewer #1: Yes

Reviewer #2: Yes

Reviewer #1: The paper presents an interesting study that is very relevant considering the rising interest in the tested methodology. I have a few small remarks:

- When introducing the case in the beginning, the real name of the perpetrator should be used. The case is very well-known, and so changing the name here seems odd. It can later be explained why the name was changed for the materials (I guess to make it more suitable to a German speaking audience).

- It would be good to include some more research on Tegenspraak, such as for instance the research done by Salet (2015)

-At the end of page 3, a lot of emphasis is placed on the role of pressure in developing the initial theory of guilt. It would be good to also place more emphases on the role of evidence or information in this initial phase.

- In the section "Mitigating Confirmation bias in police investigations", the distinction between the approach of using alternative hypotheses and focusing on falsification can be clarified more. Although related, they are not interchangeable - which can also be seen in the findings of some of the studies that are described.

- Line 96 - it should be clarified more that null effects of the intervention in these studies was largely due to a lack of bias being induced in participants (and so the intervention could not really be tested). Although it is mentioned a few lines later, it is a key aspect of these studies in relation to yours.

-Line 112 - In the Maegherman study, the instructions to come up with scenarios were not specifically related to a single suspect, and 2 suspects were suggested in the material. Participants also created scenarios about both suspects. It may also be good to consider Maegherman et al., 2021 (Accountability, where one of the conditions was focused on falsification).

-Line 234 - typo "ratings"

- For the coding, it would be helpful to include more examples. In general, the distinction between the 2 manipulations is something that remains a little unclear. How is a distinction made between suspect-focused and evidence-focused if the evidence focused still includes suspects as well? Or is the distinction based on whether multiple suspects are named? And the relation between the verification/falsification and suspect-focused and evidence focused manipulations? This is something that applies throughout the paper, although can potentially be resolved earlier on.

- The example given on line 314 is extremely clear, but it seems this would have been exceptionally clear, so other examples would be useful.

-Claiming accuracy of guilt in the case material might be tricky - even in this case, there is still professional judges who claim that there was sufficient evidence, and therefore it is not a wrongful conviction. Perhaps it would be good to clarify the use of the material truth and not the legal truth, or to rephrase the use of accuracy.

-Can the choice of investigative scenarios be related to the impression or assessment of guilt?

- Line 513 to 515 - where is the support for this? Especially considering the later acknowledged limitation that there may have been a lack of bias in the current study, but also the relation between the evidence considered/investigative measures chosen, and the final assessment of guilt not seeming clear makes this a tentative claim.

-Line 521 - How could this be done practically in a police investigation, where an idea of guilt is inherent to a suspect being identified?

-Line 587 - maybe also look at Schmittat & Englich, 2016 - expertise effect in confirmatory information processing.

In general, is there anything about the training of German police officers that is different from the Dutch that may be worth mentioning?

Reviewer #2: Thank you for the opportunity to review this manuscript. The authors examined whether encouraging falsification over verification or focusing on the evidence rather than the suspect could reduce confirmation bias in criminal investigations. This is a well written paper, an interesting topic, and includes many advanced statical analyses. However, I do have some concerns with are outlined below.

• On my initial read, I was confused by the label “exonerating evidence.” Is encouraging someone to falsify exonerating evidence not actually encouraging the verification of the initial suspect? Because, as the authors point out, this evidence is meant to be ambiguous, I would encourage them to change this label to “ambiguous additional evidence,” or something of the sort.

• I wonder why the authors chose not to include a control condition in their experimental design. It would have been valuable to understand what the officers do and think when left to their own devices. Without a control, it is difficult to understand the extent to which the debiasing effects would actually mitigate confirmation bias (because the other conditions encourage confirmation bias).

• I am concerned about the low interrater agreement rate. Why not spend more time training research assistants or revising the coding scheme to increase agreement? Particularly because these are the analyses which obtained effects, they are difficult to interpret in light of such low interrater agreement (particularly when the second coder only coded a sub sample of 20%). Also, how did the authors resolve discrepancies?

• Is it possible that the officers were familiar with the Schiedammer park murder case? Could this have affected their results? The authors refer to it as a “high profile miscarriage of justice.”

Overall, I believe that this paper has merit (and I am very impressed with their officer sample!) but the methodological concerns that I listed could undermine the intellectual contributions that the authors seek to offer.

what does this mean?). If published, this will include your full peer review and any attached files.

**Do you want your identity to be public for this peer review?** For information about this choice, including consent withdrawal, please see our Privacy Policy

Reviewer #1: No

Reviewer #2: No

While revising your submission, please upload your figure files to the Preflight Analysis and Conversion Engine (PACE) digital diagnostic tool, https://pacev2.apexcovantage.com/. PACE helps ensure that figures meet PLOS requirements. To use PACE, you must first register as a user. Registration is free. Then, login and navigate to the UPLOAD tab, where you will find detailed instructions on how to use the tool. If you encounter any issues or have any questions when using PACE, please email PLOS at figures@plos.org

---

## [Author Response · Author response to Decision Letter 1]

26 Nov 2025

We would like to thank the reviewers and the editor for their thoughtful and constructive feedback on our manuscript titled 'What Happened and What Proves You Wrong? Combating Confirmation Bias in Police Investigations Through Evidence Reconstruction and Falsification'. We carefully considered all points raised and have revised the manuscript accordingly. Below, we respond to each point one by one. Reviewer comments are bolded. In-text revisions are marked in blue.

Reviewer #1

1) When introducing the case in the beginning, the real name of the perpetrator should be used. The case is very well-known, and so changing the name here seems odd. It can later be explained why the name was changed for the materials (I guess to make it more suitable to a German speaking audience).

RESPONSE: We agree and now mention the real name - Kees B - at the first mention of the case. We clarify that in the materials, we used a pseudonym to reduce recognition of the case.

2) It would be good to include some more research on Tegenspraak, such as for instance the research done by Salet (2015)

RESPONSE: We agree and now introduce the Devil´s Advocate approach (‘Tegenspraak’) that was introduced as one of the major reforms after the erroneous conviction of kees B in the Schiedammer Parkmoord case. Here, we now reference the dissertation of Salet (2015).

3) At the end of page 3, a lot of emphasis is placed on the role of pressure in developing the initial theory of guilt. It would be good to also place more emphasis on the role of evidence or information in this initial phase.

RESPONSE: We are describing here how pressure and a focus on guilt is undesirable yet common in practice, and may bias the investigation. We agree that initial incriminating evidence is part of this process and we now elaborate our explanation on the role of evidence in this phase.

4) In the section "Mitigating Confirmation bias in police investigations", the distinction between the approach of using alternative hypotheses and focusing on falsification can be clarified more. Although related, they are not interchangeable - which can also be seen in the findings of some of the studies that are described.

RESPONSE: We appreciate this critical insight, as making clear the distinction between falsification (trying to prove a hypothesis wrong by searching for evidence that contradicts it) and considering alternative hypotheses. We have now clarified this by rewriting this section.

5) Line 96 - it should be clarified more that null effects of the intervention in these studies was largely due to a lack of bias being induced in participants (and so the intervention could not really be tested). Although it is mentioned a few lines later, it is a key aspect of these studies in relation to yours.

RESPONSE: We indeed mentioned ‘failed or insufficient manipulations’ as a key drawback of some previous studies. We now moved this argument forward and made it more explicit.

6) Line 112 - In the Maegherman study, the instructions to come up with scenarios were not specifically related to a single suspect, and 2 suspects were suggested in the material. Participants also created scenarios about both suspects. It may also be good to consider Maegherman et al., 2021 (Accountability, where one of the conditions was focused on falsification).

RESPONSE: We appreciate the reviewer’s careful reading and have clarified our reasoning. While the Maegherman case materials subtly implied an alternative explanation involving the victim’s mistress, the study centered on a single main suspect (Sabine). The design and analyses primarily focused on judgments regarding Sabine’s guilt. We have revised the relevant section to more accurately reflect this. We also appreciate the suggested study by Maegherman and now include it in the present manuscript.

7) Line 234 - typo "ratings"

RESPONSE: Thank you, corrected.

8) For the coding, it would be helpful to include more examples. In general, the distinction between the 2 manipulations is something that remains a little unclear. How is a distinction made between suspect-focused and evidence-focused if the evidence focused still includes suspects as well? Or is the distinction based on whether multiple suspects are named? And the relation between the verification/falsification and suspect-focused and evidence focused manipulations? This is something that applies throughout the paper, although can potentially be resolved earlier on.

RESPONSE: The distinction between evidence focus and suspect focus is indeed not always easy, for instance when the mentioned evidence is related to a suspect. This reflects real world investigative practices and is the primary reason for choosing a continuous scale ranging from fully suspect-focused to fully evidence-focused (rather than a binary judgment). The reasoning strategy (verification- vs. falsification-focused) is distinct from the investigate focus (suspect- vs. evidence-focused): It is about searching for evidence that fits one's theory (verification) versus evidence that contradicts it (falsification). We have clarified these distinctions in the text, and added more examples to the coding scheme. We have also added a rationale for some examples for added clarity.

9) The example given on line 314 is extremely clear, but it seems this would have been exceptionally clear, so other examples would be useful.

RESPONSE: We are happy to see this example clearly illustrates our coding of the next investigative steps. Per reviewer request, we have added another example that may be less straightforward to code.

10) Claiming accuracy of guilt in the case material might be tricky - even in this case, there is still professional judges who claim that there was sufficient evidence, and therefore it is not a wrongful conviction. Perhaps it would be good to clarify the use of the material truth and not the legal truth, or to rephrase the use of accuracy.

RESPONSE: We agree and now clarify that with accuracy we refer to how closely participants’ conclusions align with the material truth of the case as presented in the study materials, not the legal truth as determined by the courts.

11) Can the choice of investigative scenarios be related to the impression or assessment of guilt?

RESPONSE: Participants did not choose between investigative scenario types: These were experimentally assigned through task instructions. We therefore interpret the reviewer’s question as referring to the content of the scenarios participants produced within the task. We agree that the scenarios constructed could, in principle, reflect participants’ impression of guilt. However, our coding scheme explicitly accounted for this: when participants produced multiple scenarios centered on the same suspect, the response was not coded as fully evidence-focused but rather as mixed (see Appendix A). This ensured that suspect-centered reasoning was distinguished from genuinely evidence-based reconstruction. The case materials and information were identical across conditions, and our manipulation checks and analyses showed no reliable effect of focus or strategy on guilt ratings, reducing concern that instructional framing itself biased perceptions of guilt.

12) Line 513 to 515 - where is the support for this? Especially considering the later acknowledged limitation that there may have been a lack of bias in the current study, but also the relation between the evidence considered/investigative measures chosen, and the final assessment of guilt not seeming clear makes this a tentative claim.

RESPONSE: While acknowledging the limitations of our study, we think that our conclusion that ‘The present findings suggest that while final judgements in investigative processes may be relatively resistant to change, early-stage investigative processes may be more receptive to debiasing efforts’ is justified by the data, as reported in the section ‘Secondary Outcome: Next Investigative Steps’, page 17-18. We now added a reference to Figures 2-3 (S2-S3 Figs). We also readily acknowledge the study limitations and use language that shows some uncertainty (i.e., ‘suggests’, ‘may’).

13) Line 521 - How could this be done practically in a police investigation, where an idea of guilt is inherent to a suspect being identified?

RESPONSE: We now add the Scenario Reconstruction Method as an example of how to keep early-stage investigative steps bias-free and evidence-focused. This method is regularly taught and applied in criminal investigations in The Netherlands. We now mention this method as a good example, and refer the reader to Epskamp-Dudink & Winter (2019).

14) Line 587 - maybe also look at Schmittat & Englich, 2016 - expertise effect in confirmatory information processing.

RESPONSE: We appreciate the suggestion and now add a reference to Schmittat & Englich (2016) who also found ‘that legal experts are not protected against confirmatory information processing’. We have added the reference.

15) In general, is there anything about the training of German police officers that is different from the Dutch that may be worth mentioning?

RESPONSE: We are not sufficiently familiar with the training of Dutch vs German police officers to speculate on putative differences. As far as we know, Dutch and German legal systems are rather similar.

Reviewer #2

16) Thank you for the opportunity to review this manuscript. The authors examined whether encouraging falsification over verification or focusing on the evidence rather than the suspect could reduce confirmation bias in criminal investigations. This is a well written paper, an interesting topic, and includes many advanced statical analyses. However, I do have some concerns with are outlined below.

RESPONSE: We appreciate the encouraging feedback as well as your helpful concerns and recommendations.

17) On my initial read, I was confused by the label “exonerating evidence.” Is encouraging someone to falsify exonerating evidence not actually encouraging the verification of the initial suspect? Because, as the authors point out, this evidence is meant to be ambiguous, I would encourage them to change this label to “ambiguous additional evidence,” or something of the sort.

RESPONSE: On page 10, line 247 we had a header ‘Exonerating evidence’, but per reviewer suggestion, we have now changed that label to 'ambiguous additional evidence' throughout the manuscript.

18) I wonder why the authors chose not to include a control condition in their experimental design. It would have been valuable to understand what the officers do and think when left to their own devices. Without a control, it is difficult to understand the extent to which the debiasing effects would actually mitigate confirmation bias (because the other conditions encourage confirmation bias).

RESPONSE: We chose to carefully pretest the biasing nature of the vignette in pilot studies. This allowed us to maximally power our experimental conditions in the full study. But we acknowledge the lack of a control condition as an important limitation.

19) I am concerned about the low interrater agreement rate. Why not spend more time training research assistants or revising the coding scheme to increase agreement? Particularly because these are the analyses which obtained effects, they are difficult to interpret in light of such low interrater agreement (particularly when the second coder only coded a sub sample of 20%). Also, how did the authors resolve discrepancies?

RESPONSE: We developed an elaborate coding scheme (see Appendix A) with several examples on how to code. Still, the data were sometimes difficult to code and this resulted in discrepancies between coders. Disagreements were resolved through discussion in the pilot study.

For the focus manipulation check, inter-rater agreement was moderate to substantial. For the strategy manipulation check, inter-rater agreement was fair to moderate. For the next investigative steps, inter-rater agreement was fair to moderate.

We also acknowledge that the inter-rater reliability could use improvement. Like Reviewer 2 suggests, we recommend to further improve the coding scheme, and a more extensive training for future research.

20) Is it possible that the officers were familiar with the Schiedammer park murder case? Could this have affected their results? The authors refer to it as a “high profile miscarriage of justice.”

RESPONSE: The Schiedammer Park murder case is well known in the Netherlands. To minimize any risk of familiarity, we recruited officers from another country (Germany), who would be unlikely to recognize the case. We also changed all identifying details (including names, ages, locations, and contextual factors) to prevent participants from drawing a connection to the original case. For these reasons, we did not consider it necessary to explicitly ask participants about their familiarity. However, we did consult with teachers at the police academy (through which we collected most of our data) whether they covered cases from other countries in their curriculum, and if students were likely to be familiar with the Schiedammer park murder. Neither was the case.

21) Journal requirements: Please ensure that your manuscript meets PLOS ONE's style requirements, including those for file naming. The PLOS ONE style templates can be found at https://journals.plos.org/plosone/s/file?id=wjVg/PLOSOne_formatting_sample_main_body.pdf and https://journals.plos.org/plosone/s/file?id=ba62/PLOSOne_formatting_sample_title_authors_affiliations.pdf

RESPONSE: Thank you, we have adapted the manuscript accordingly.

22) Thank you for uploading your study's underlying data set. Unfortunately, the repository you have noted in your Data Availability statement does not qualify as an acceptable data repository according to PLOS's standards.At this time, please upload the minimal data set necessary to replicate your study's findings to a stable, public repository (such as figshare or Dryad) and provide us with the relevant URLs, DOIs, or accession numbers that may be used to access these data. For a list of recommended repositories and additional information on PLOS standards for data deposition, please see https://journals.plos.org/plosone/s/recommended-repositories.

RESPONSE: Thank you, we have now uploaded the data to figshare: 10.6084/m9.figshare.30518270

23) Please include your full ethics statement in the ‘Methods’ section of your manuscript file. In your statement, please include the full name of the IRB or ethics committee who approved or waived your study, as well as whether or not you obtained informed written or verbal consent. If consent was waived for your study, please include this information in your statement as well.

RESPONSE: We have now included our full ethics statement in the ‘Methods’ section of our manuscript file, including the full name of the ethics committee and that we obtained digital informed consent.

24) Please include a separate caption for each figure in your manuscript.

RESPONSE: Done.

25) Please include captions for your Supporting Information files at the end of your manuscript, and update any in-text citations to match accordingly. Please see our Supporting Information guidelines for more information: http://journals.plos.org/plosone/s/supporting-information.

RESPONSE: Thank you, done.

---

## [Editor Report · Decision Letter 1]

3 Dec 2025

What happened and what proves you wrong? Combatting confirmation bias in police investigations through evidence reconstruction and falsification

PONE-D-25-29848R1

Dear Dr. Verschuere,

We’re pleased to inform you that your manuscript has been judged scientifically suitable for publication and will be formally accepted for publication once it meets all outstanding technical requirements.

Kind regards,

Bartosz Wojciech Wojciechowski, Ph.D.

Academic Editor

PLOS ONE
---

## [Editor Report · Acceptance letter]

PONE-D-25-29848R1

PLOS One

Dear Dr. Verschuere,

I'm pleased to inform you that your manuscript has been deemed suitable for publication in PLOS One. Congratulations! Your manuscript is now being handed over to our production team.

Kind regards,

on behalf of

Dr. Bartosz Wojciech Wojciechowski

Academic Editor

PLOS One